# Piezotronic Antimony Sulphoiodide/Polyvinylidene Composite for Strain-Sensing and Energy-Harvesting Applications

**DOI:** 10.3390/s23187855

**Published:** 2023-09-13

**Authors:** Jakub Jała, Bartłomiej Nowacki, Bartłomiej Toroń

**Affiliations:** 1Department of Materials Technologies, Faculty of Materials Engineering, Joint Doctoral School, Silesian University of Technology, Krasińskiego 8, 40-019 Katowice, Poland; jakub.jala@polsl.pl; 2Department of Industrial Informatics, Faculty of Materials Engineering, Joint Doctoral School, Silesian University of Technology, Krasińskiego 8, 40-019 Katowice, Poland; bartlomiej.nowacki@polsl.pl; 3Institute of Physics—Center for Science and Education, Silesian University of Technology, Krasińskiego 8, 40-019 Katowice, Poland

**Keywords:** piezotronics, composites, functional materials, chalcohalides, nanowires, flexible electronics

## Abstract

This study investigates the piezoelectric and piezotronic properties of a novel composite material comprising polyvinylidene fluoride (PVDF) and antimony sulphoiodide (SbSI) nanowires. The material preparation method is detailed, showcasing its simplicity and reproducibility. The material’s electrical resistivity, piezoelectric response, and energy-harvesting capabilities are systematically analyzed under various deflection conditions and excitation frequencies. The piezoelectric response is characterized by the generation of charge carriers in the material due to mechanical strain, resulting in voltage output. The fundamental phenomena of charge generation, along with their influence on the material’s resistivity, are proposed. Dynamic strain testing reveals the composite’s potential as a piezoelectric nanogenerator (PENG), converting mechanical energy into electrical energy. Comparative analyses highlight the composite’s power density advantages, thereby demonstrating its potential for energy-harvesting applications. This research provides insights into the interplay between piezoelectric and piezotronic phenomena in nanocomposites and their applicability in energy-harvesting devices.

## 1. Introduction

The continuous development and miniaturization of electronics have led to improvements in almost all branches of industry, but it has also brought forth unique and increasing challenges. The demand for more precise and reliable sensors is continuously growing. Since the term “Microelectromechanical System” (MEMS) was coined in the 1980s, MEMSs have been developed and currently serve various purposes, including as accelerometers [1], strain sensors [2], optical switches [3], oscillators [4], and many more, offering a wide range of applications that provide ample opportunities for further improvements and advancements. This work focuses on strain-sensing devices, exploring their modern downscaling and specific requirements.

Strain sensing typically involves the use of one of three mechanisms: the piezoelectric effect, piezoresistivity, and capacitance. Conventional capacitive sensors offer accuracy across a wide range of pressures and are resistant to temperature changes, but their significant drawback is the nonlinearity of output and high sensitivity to vibrations [5]. On the other hand, piezoresistive sensors are more resilient to vibrations and mechanically stable, but their results are susceptible to temperature changes [6]. Piezoelectric sensors offer high sensitivity and output, enabling the detection of even subtle movements and acoustic waves [7,8]. However, they are susceptible to temperature variations and are primarily suited for dynamic measurements [9]. To address some of these disadvantages associated with these sensor types, researchers have explored the concept of piezotronic strain sensors, which may offer potential solutions.

Piezotronics, a relatively recent field, traces its origins back to 2006, when Wang and his co-authors pioneered the development of a field-effect transistor based on a single piezoelectric ZnO nanowire [10]. The fundamental principles of piezotronics at the nanoscale were introduced by Z.L. Wang a year later [11]. Piezotronics is an interdisciplinary branch of science that couples piezoelectric and semiconductor properties in certain materials. In piezoelectric materials, mechanical stress induces the generation of electric charges, while in semiconductors, mechanical deformation influences the flow of electric current. The key concept in piezotronics is the piezopotential, which refers to the potential difference created in the material due to mechanical deformation. Consequently, this piezopotential effect shapes the band structure and alters carrier dynamics, thus providing precise control over electronic transport within piezotronic devices.

Based on these principles, numerous devices and piezotronics applications have been developed. Initially, strain sensors emerged, capable of detecting both dynamic and static strain, surpassing the functionality of conventional piezoelectric sensors limited to dynamic strain sensing. Notably, flexible devices [12], pressure and vibration sensors [13], nanogenerators [14], field-effect transistors [15], and piezophototronic devices combining photonic excitation with piezotronics and photocatalytic applications [16,17] have been achieved. Additionally, advances have been made in electromechanical memory [18] and synapses [19], unveiling further possibilities for these technologies.

Among the various piezotronics-viable materials, extensive research has been focused on ZnO [20,21], and there has also been some interest in nitrides (GaN, InN) [22], tellurides (HgTe, CdTe) [15], 2D layers of transition metal dichalcogenides (TMDs), e.g., MoS_2_ [23], and perovskites [24]. However, a group of materials that has been somewhat overlooked but holds significant promise for piezotronics applications is chalcohalides. Within the ternary chalcohalides compounds, SbSI, SbSeI, and BiSI stand out as notable materials with diverse applications, including gas sensors [25], nanogenerators [26], photodetectors [27], smart textiles [28], and strain sensors [29]. Among these ternary chalcohalides, SbSI emerges as the most promising piezoelectric material. It exhibits a high piezoelectric coefficient (d_33_ = 1 nC/N) [30], surpassing commercially used materials like KNN, PZT, and BTO. Furthermore, its Curie temperature (approx. 295K [31]) enables room-temperature applications, and it boasts excellent pyroelectric properties [32], while also demonstrating highly encouraging results in photocatalysis [33].

Over the last decade, there has been renewed interest in antimony sulphoiodide due to its unique properties. Notably, the relatively straightforward production of SbSI nanowires through sonochemical synthesis [31] enhances its potential for multifunctional strain sensors, energy harvesters, and piezophototronic devices, opening new possibilities for advanced applications. Moreover, the combination of antimony sulphoiodide with other materials in the form of nanocomposites shows promise for further improving efficiency and capabilities through a synergistic effect. Applying different reinforcing materials [34,35] and applying an external electric field [36] lead to a composite with different mechanical and piezoelectric properties.

In this study, we present a novel composite material comprising semiconductor piezoelectric SbSI nanowires, embedded within a flexible piezoelectric PVDF polymer matrix. This material is investigated in piezotronic strain-sensing applications for the first time. The exceptional piezoelectric parameters exhibited by SbSI, surpassing those of comparable conventional semiconducting piezoelectrics, make it an ideal choice for such purposes. The incorporation of a PVDF matrix, being a piezoelectric polymer, ensures a flexible and accommodating matrix that minimally affects the composite’s sensing capabilities, unlike more commonly used polymers. The size of SbSI nanowires, ranging from several micrometers in length to tens or dozens of nanometers in lateral dimensions [37], facilitates the easy scalability of the produced devices to meet the requirements of MEMSs. The promising performance of ternary chalcohalides, such as SbSI, in piezotronics, opens the door to explore more advanced applications. This includes the development of more complex devices and systems, with potential applications extending to sensing synapses, memory, and logic devices, among others. This will lead to the development of more complex systems and devices, paving the way for the development of modern architecture-based memory and computing devices. Further research in this field could unveil even more possibilities, propelling the utilization of ternary chalcohalides in the domain of piezotronics. This study investigates a thick-film material, which has been fabricated using a novel method. This approach differs significantly from the SbSI-PVDF system, which was previously fabricated by the method of spinning fibers as presented in [28].

## 2. Sample Preparation

SbSI nanowires were synthesized using the sonochemical method. Stoichiometric amounts of pure elements—Sb, S, and I—were combined in ethanol within a closed container. Subsequently, the mixtures underwent ultrasonic treatment in an Ulsonix Proclean ultrasonic washer (Ulsonix, Germany) for a duration of 2 h. The sonification process resulted in the formation of a gel, which was then subjected to centrifugation and washed ten times with ethanol to remove impurities. The purified ethanogel was subsequently dried at room temperature under reduced pressure. The resulting nanowires were then ready for use in nanocomposite fabrication. For a more in-depth description of the fabrication process and the properties of SbSI nanowires, please refer to the literature [31]. PVDF was dissolved in DMF (20:80 ratio) at a temperature of 60 °C with continuous stirring for 4 h. Once PVDF was completely dissolved, SbSI was added to the solution, which was then placed on a hot plate with a magnetic stirrer. Four different samples were prepared: one reference sample containing only PVDF, and three composite samples with 5, 10, and 15 weight % of SbSI, respectively.

The samples were prepared using a simple and scalable casting method. After mixing SbSI with PVDF, the resulting mixtures were cast into rectangular silicone molds, heated up to 80 °C, and left for 12 h to remove the DMF solvent. Afterwards, both sides of each sample were coated with conductive carbon paint (Bare Conductive, United Kingdom) forming electrodes. Copper wires were placed on the electrodes before the conductive paint dried, establishing the necessary electrical connections. The sample area, i.e., the area between the electrodes, was 2 cm^2^, and the thickness was measured to be 1 mm. The dimensions of the electrode area and sample thickness were measured using calipers, with a limiting uncertainty of 0.05 mm. A visual representation of the sample preparation method is presented in Figure 1.

## 3. Testing and Measurement Procedures

SEM (scanning electron microscope) and EDS (energy-dispersive X-ray spectroscopy) examination was performed on the samples to ensure proper dispersion of SbSI nanowires throughout the sample. The Phenom Pro X scanning electron microscope equipped with EDS and manufactured by Thermo Fisher Scientific (Waltham, MA, USA) was employed for this purpose. The surfaces of the samples were investigated through SEM and EDS studies. The grain size of PVDF was analyzed using the specialized ParticleMetric program within the Phenom ProSuite Software.

The samples were initially subjected to electric poling using an external electric field. The poling process was conducted with a voltage of 350 V (i.e., an electric field of E = 3.5 kV/cm) applied to the samples for a duration of one hour. For this purpose, a Keithley 2410-C SourceMeter (Cleveland, OH, USA) was utilized. The voltage value used in poling was based on the literature [38].

All measurements were conducted at room temperature and under ambient conditions. For static testing, 3D-printed PLA (Prusament by Prusa, Prague, Czech Republic) arc-shaped elements with curvatures of 15, 30, and 60 degrees were prepared. During the measurements, the samples were positioned between these elements, resulting in the respective deflections. During static testing, I-V dependencies were recorded using a Keithley 6517B electrometer (Cleveland, OH, USA), which was controlled by a PC running a program written in LabVIEW 2015 software developed by National Instruments (Austin, TX, USA). A schematic presentation of measurements is shown in Figure 2.

For dynamic testing, acoustic wave stimulation was applied using a 2008 speaker manufactured by Visaton (Germany). The speaker was controlled by a Metex MXG-9802A Function Generator (South Korea). The sine and rectangular waveforms generated by the function generator were further amplified using a solid-state TDA7496-based power amplifier manufactured by STMicroelectronics (France/Italy). Data from the dynamic measurements, including the sample’s response, were acquired using a DSOX3104T oscilloscope manufactured by Keysight Technologies (Santa Rosa, CA, USA). During the examination, the specimens were affixed and positioned above the loudspeaker to facilitate acoustic pressure measurements. The microcontroller equipped with the ST MP34DT06JTR microphone was employed to accurately capture the acoustic pressure produced by the speaker. Specifically, the sound pressure level (SPL) was quantified at a frequency of 1 kHz, utilizing a sine wave, yielding an estimated value of 0.02 N/mm^2^. Subsequently, employing the frequency response data furnished by the speaker’s manufacturer, the recalibration of pressure measurements was performed for various frequencies under consideration. Within the designated measurement frequencies, the analysis revealed recalibrated pressure values of 0.0177 N/mm^2^ at 100 Hz, 0.0207 N/mm^2^ at 200 Hz, and a consistent 0.02 N/mm^2^ at 400 Hz frequency. The acquired data were then analyzed and presented using OriginPro 8.5 software by OriginLab (Northampton, MA, USA).

## 4. Results and Discussion

### 4.1. SEM Examination

SbSI nanowires exhibit a strong tendency to agglomerate within the PVDF matrix [28]. Unfortunately, this agglomeration significantly deteriorates the piezoelectric properties of the nanocomposites [39,40,41]. The potential presence of large agglomerates could lead to non-uniform piezoelectric performance across the sample volume. This non-uniformity could significantly reduce the overall suitability of the device for its intended applications. Even when the composite contains only a small amount of SbSI, some agglomerates still can form due to the inherent agglomeration propensity of the nanowires. To address this issue, SEM tests were conducted for various percentages of SbSI content in the nanocomposite, and the results are presented in Figure 3.

Appropriate deagglomeration of the nanowires was observed, resulting from ultrasound application during material preparation. However, in cases in which a higher percentage of SbSI is added, it is advisable to consider incorporating a solvent or adjusting the mixing time. These approaches have been suggested for enhancing the properties of many nanocomposites [42,43], though it is worth underlining that all process parameters remained constant in this study.

The random orientation of the nanowires indicates the formation of a 0-3 type nanocomposite [44], which might have implications for the efficiency and effects of ferroelectric polling [45,46]. EDS measurement (Figure 3e) revealed no significant changes or reactions of the nanowire filler during the nanocomposite preparation. The atomic ratio of antimony, sulfur, and iodine confirms the stoichiometric composition of SbSI, consistent with prior research on this material [31]. For composite compositions with different concentrations of SbSI, a similar content ratio was obtained. Additionally, the preparation conditions for all films, except for the varying SbSI nanowire content, remained consistent. Therefore, they should have undergone the same DMF evaporation process, which has a significant influence on the grain size of PVDF. Furthermore, the grain size of PVDF is also affected by the concentration of nanowires present. Previous research has shown that the quantity of nanofillers in the composite directly influences the size of PVDF grains [47]. In case of this research, the influence of nanowires on grain size is slightly different than in [47]. Notable changes were observed specifically in the case of the highest SbSI content (Figure 3f), in which the largest grains of PVDF can be seen for the most amount of nanofiller. This might be attributed to alterations in viscosity and surface tension dynamics caused by the increased nanowire concentration.

### 4.2. Characterization of Samples under Static Deformation

Each sample underwent examination at four different deflection angles of 0°, 15°, 30°, and 60°. Samples were placed between pre-prepared arch-shaped elements to achieve the desired deflection angle. I-V characteristics were then measured for each sample under each deflection, with a bias voltage range of −5 V to 5 V (a range chosen due to industry standards for microcontrollers).

Figure 4 and Figure 5 illustrate a comparison of the I-V plots on a single graph for different samples under the same deflection angle and under various deflection angles for the same sample, respectively, aiming to underscore the influence of each of these factors.

Linear functions were fitted to the measured data using the least-squares method. In the achieved data points, differences between the measured samples under varying deflections were found. However, these nonlinearities are so small that straight lines were still fitted to investigate the influence of deflection angle and SbSI content on the electrical resistivity of the material. The fitting parameters of the I-V dependencies are further presented in Table 1.

Figure 4 displays the I-V curves for different weight contents of SbSI in the nanocomposite under the same deflection. As the amount of SbSI nanowires increases, the measured line slope decreases, indicating an increase in electrical resistance. This trend can be observed in all tested deflections. The difference is visible for undeflected samples (Figure 4a), in which neat PVDF (red line) shows slightly higher electrical resistance than samples with 5 and 10 wt. % of SbSI.

In Figure 5, the I-V curves for investigated nanocomposite samples with different weight contents of SbSI under various deflections are presented. For composite samples (Figure 5b–d), an increase in deflection leads to a decrease in the line slope, resulting in an increase in resistance. However, the opposite relationship is observed for PVDF (Figure 5a). Moreover, the measured characteristics exhibit slight nonlinear aspects, particularly evident in Figure 5b,c, in which points for higher applied voltages deviate from the fitted line.

Notably, the samples with 5% and 10% SbSI filler display I-V characteristics under 30° and 60° deflections, respectively, that resemble the responses of piezotronic devices described in the literature [48,49]. While conductivity modulation by mechanical strain has been achieved in these samples, it appears to be relatively lower than that observed in non-composite piezotronic materials. Unlike the typical Schottky junctions found in most piezotronic devices, the contacts in the examined samples have metal–insulator–semiconductor (MIS) structures corresponding to an electrode–PVDF–SbSI structure in this case. The MIS structure finds applications in advanced electronics, like memory devices [50]. This MIS setup allows for different behavior and characteristics compared to Schottky junctions, and it might be contributing to the observed electrical resistance changes and piezotronic effects in the nanocomposite under varying deflections. The MIS system seems to function similarly to a gate modulated by strain. The presence of an insulator in the investigated systems may inhibit the full piezotronic effect, but as mentioned earlier, the desired effect is still discernible. With further research, this system could be optimized to develop high-accuracy devices of this type.

To examine the influence of deflection and weight content of SbSI on the electrical resistivity of the nanocomposite, the following calculations were performed. First of all, it can be seen that the “b” coefficient in Table 1 is nearly zero within the uncertainty limit and can thus be omitted in the calculation. Therefore, electrical resistance (R) was determined as the reciprocal of the “a” coefficient from the linearly fitted I-V curves:(1)$$R=\frac{1}{a}$$
where “a” represents the slope of the straight-line fit. Subsequently, electric resistivity (ρ) was calculated, taking into account the sample dimensions:(2)$$\rho =R\frac{A}{d}$$
where “A” is the sample’s active area, “d” is the sample thickness, and “R” is the sample’s resistance.

For the calculation of uncertainties in electrical resistance and resistivity, the uncertainty propagation law was applied, employing the following formulas:(3)$$u\left(R\right)=\sqrt{{\left(\frac{\partial R}{\partial a}\cdot u(a)\right)}^{2}}=\sqrt{{\left(-\frac{1}{a^{2}}\cdot u(a)\right)}^{2}}$$
(4)$$u\left(\rho \right)=\sqrt{{\left(\frac{\partial \rho }{\partial R}\cdot u(R)\right)}^{2}+{\left(\frac{\partial \rho }{\partial A}\cdot u(A)\right)}^{2}+{\left(\frac{\partial \rho }{\partial d}\cdot u(l)\right)}^{2}}=\sqrt{{\left(\frac{A}{d}\cdot u(R)\right)}^{2}+{\left(\frac{R}{d}\cdot u(A)\right)}^{2}+{\left(-\frac{\mathrm{RA}}{d^{2}}\cdot u(d)\right)}^{2}}$$
where “u(a)”, “u(A)”, and “u(d)” are the uncertainties of the slope, sample area, and sample thickness, respectively. The sample area uncertainty was also calculated using the uncertainty propagation law, considering the electrode dimensions measured with calipers. Table 1 presents the fitting parameters of the I-V dependencies and the results of these calculations.

The calculated electrical resistivity as a function of deflection and weight content of SbSI is presented in Figure 6.

The values of PVDF resistivity are reported in the range of 10^11^ Ω·cm [51], whereas antimony sulphoiodide exhibits a resistivity of approximately 1 × 10^9^ Ω·cm [37]. The results in this work show that the resistivity generally increases with the weight content of SbSI (Figure 6b). We suggest the reason behind this to be the trapping of charge carriers on the boundaries of antimony sulphoiodide nanowires within the composite. Further discussion of this seemingly anomalous influence of SbSI is presented further in the text and explained in the description of Figure 7. 

Regarding SbSI composites, during strain and deflection, the displacement of randomly oriented nanowires indicates a clear tendency for increasing resistivity, exhibiting what is commonly referred to as the positive pressure coefficient of resistance (PPCR), which is a common occurrence in materials with high aspect ratio fillers, such as nanowires or nanotubes [52]. 

The electrical resistivity of the sample changes with varying degrees of deflection. For the neat PVDF sample, an increase in the degree of deflection from 0 to 60 degrees leads to a decrease in resistance, from approximately 393 to 294 GΩ·cm, as depicted in Figure 6a. However, the relation between resistivity and deflection is reversed for composite samples. For samples containing 5 and 10 weight percent of SbSI, their resistivity increases by 70 and 98 GΩ·cm, respectively, corresponding to about a 21% and 26% increase when going from 0 to 60 degrees of deflection. On the other hand, the composite with the highest concentration of antimony sulphoiodide filler (15 wt. %) experiences the most drastic increase in resistivity under strain, rising from 1087 to 4155 GΩ·cm, which is a notable four-fold increase compared to samples with 5% and 10% wt. filler (Figure 6). It is expected that the resistivity will not increase to infinity but rather remain constant for significantly higher deflection angles. Consequently, an empirical relationship with a horizontal asymptote has been fitted to the electrical resistivity plotted against the sample deflection angle (Figure 6a). The equation of the fitted function is as follows:(5)$${\rho (\alpha )=a}_{1}-b_{1}\;\cdot {\;c}_{1}^{\alpha }$$
where a_1_, b_1_, and c_1_, are the fitting coefficients. The fitting procedure was conducted using OriginPro 8.5 software, taking into account the uncertainties in resistivity. The values of the fitted coefficients are presented in Table 2.

The fitted coefficients of the nanocomposite samples change monotonously with the SbSI filler content, except for the pure PVDF sample, which exhibits a reverse trend in resistivity vs. deflection changes. The resulting functions show a determination coefficient of nearly 1 for both neat PVDF and PVDF with 15%wt. SbSI. The ρ(α) dependencies become more linear with a decrease in SbSI content, resulting in the c_1_ coefficient approaching 1. The c_1_ coefficient signifies the rate of resistivity changes concerning deflection, with a smaller c_1_ value indicating a higher resistivity change rate. Together, the b_1_ and c_1_ coefficients describe how the sample resistivity changes in response to deflection. The sign of the b_1_ coefficient determines the direction of resistivity changes, where negative b_1_ for a pure PVDF sample indicates a decrease in resistivity with deflection, while a positive value of b_1_ for composite samples indicates an increase in resistivity with an increase in deflection angle. As c_1_ < 1, the horizontal asymptote for function (5) is reached as follows:(6)$$\underset{\alpha \to \infty }{\lim}\rho \left(\alpha \right){=a}_{1}$$

Therefore, the a_1_ coefficients represent the limit resistivity for high deflection angles, which can be envisioned as the resistivity of a rolled sample.

One can see that the electrical resistivity plotted vs. weighted content of SbSI nanowires exhibits the exponential growth relation in the examined content of SbSI for all deflection angles. Therefore, the appropriate functions have been fitted:(7)$$\rho \left(\mathrm{wt}\right){=\rho }_{0}{+a}_{2}\;\cdot \;\exp\left(\frac{\mathrm{wt}}{b_{2}}\right)$$
where ρ_0_, a_2_, and b_2_, are the fitting coefficients, corresponding to resistivity for neat PVDF, amplitude of resistivity changes, and growth constant, respectively. As stated previously, the fitting procedure was conducted using OriginPro 8.5 software, taking into account the uncertainties in resistivity. The values of the fitted coefficients are presented in Table 3.

One can observe a slight decrease in resistivity with an increase in deflection for pure PVDF (ρ_0_). However, this relationship is reversed for the nanocomposite with 15%wt. of SbSI (Figure 6b). Additionally, both the amplitude of resistivity changes and the growth constant tend to be constant values with an increase in deflection. This confirms the previous assumption that the sample resistance will not increase to infinity but will remain constant for high deflection angles. Moreover, it must be noted that the nanocomposite resistivity will not reach infinite values with an increase in SbSI content since the maximum SbSI composition is limited, as composites with high SbSI content will lose their flexibility. 

It is essential to consider that the proposed dependencies between resistivity, deflection, and filler content in piezotronic nanocomposites are empirical due to the absence of fundamental principles (due to the inherent randomness and complexity of such a system). Nonetheless, these behaviors can be understood and explained based on the piezoelectric phenomena of the nanocomposite filler, such as SbSI, in combination with the matrix material, i.e., PVDF. By analyzing the interactions between the filler and the matrix, one can gain insights into the observed changes in resistivity and their relation to deflection and filler content. Thus, while the dependencies may lack a theoretical foundation, the piezoelectric properties of the nanocomposite components play a significant role in explaining these empirical trends. In this case, the PPCR effect of the nanocomposite can be explained based on the phenomena depicted in Figure 7.

Figure 7 presents a simplified illustration of the effects within the examined composite. On one hand, the polymer material, PVDF, comprises relatively flexible polymer structures. As the deflection increases, the polymer chains elongate and shift into more favorable positions for electrical conductivity, enhancing carrier mobility. This behavior of PVDF resistivity under strain aligns with previous works in the literature, in which PVDF is utilized as a piezoresistive strain sensor [53,54], as shown in the neat PVDF sample (Figure 6a).

On the other hand, when mechanical strain is applied to the PVDF/SbSI nanocomposite, piezopotential is induced in SbSI nanowires, as depicted in Figure 7b. Due to the presence of piezopotential, the nanowires can be treated as electric dipoles (Figure 7b(II)), and thus, each of them attracts charge carriers towards the opposite polarity on the crystal surface (Figure 7b(III)). The dipole polarization is reversed when the nanowires are compressed or elongated (Figure 7b,c). Consequently, these dipoles attract charge carriers within the composite, trapping them on the surface of the SbSI nanowires, thus reducing the number of carriers available for electrical conduction. This phenomenon hampers electrical current conduction by limiting the charge carriers available on the surface of SbSI nanowires. Moreover, during nanocomposite deflection, nanowires on the inner plane of curvature can be compressed, while those on the outer plane of curvature can be elongated simultaneously.

Antimony sulphoiodide generates notable amounts of carriers, demonstrates significant advantages in piezoelectric performance, and serves as the main source of charge generation in the nanocomposite when subjected to dynamic strain, as shown in the next section, but during static deflection, it counteracts conductivity due to the dipoles. This effect becomes more significant with higher amounts of SbSI filler in the composite. 

### 4.3. Dynamic Strain Testing

Experimental dynamic tests were conducted to investigate the potential application of the presented nanocomposite as piezoelectric nanogenerators (PENGs). The energy-harvesting mechanism of the PVDF/SbSI nanocomposite follows a similar concept illustrated in Figure 7. In this case, dynamic changes in the excitation, such as sound pressure here, induce stress on the surfaces of the PENG. This stress is then transmitted to the piezoelectric SbSI nanowires through the PVDF matrix. When the strained composite is connected to an external electric load, electric charges are generated at the ends of the SbSI nanowires (Figure 7b,c), leading to a net potential difference across the PENG device’s carbon electrodes. Consequently, free electrons within the external load move from one side to the other, balancing the potential of the PENG sample and establishing a new equilibrium. The resulting current in the load arises from this transient flow of electrons. To maintain the continuous output power of the PENG device and achieve an alternating flow of electrons, dynamic stress is continually applied across the composite, causing the potential to change continuously. 

The response of the samples was carefully investigated in the presence of both sine and square wave excitations, spanning frequencies of 0.1, 0.2, 0.4, and 1 kHz. Throughout this investigation, the sound power was accurately maintained at a consistent level across all frequencies under scrutiny. The outcomes of this study are depicted in Figure 8 and Figure 9. These graphical representations illustrate the voltage waveforms characterizing the piezoelectric reactions to the acoustic wave stimuli. Particularly, Figure 8 highlights the voltage responses exhibited by samples containing varying concentrations of SbSI when subjected to sine acoustic wave stimuli across diverse frequencies.

The peak-to-peak voltage (U_p-p_) can be computed based on the subsequent 20 peaks of the acquired signals, which correspond to the excitation frequencies. The associated uncertainty is derived as the standard deviation. This value, divided by the force exerted by the sine wave, facilitates the calculation of the peak-to-peak voltage under the applied force, denoted as “U_p-pF_”. The expression for this relationship is articulated as follows:(8)$$U_{p\mathrm{-pF}}=\frac{U_{\mathrm{p-}p}}{F}=\frac{U_{\mathrm{p-}p}}{\mathrm{pS}}$$
where “U_p-p_” signifies the peak-to-peak voltage and “F” denotes the force originating from the sine wave excitation, determined from the previously estimated pressures “p” of 0.0177 N/mm^2^ at 100 Hz, 0.0207 N/mm^2^ at 200 Hz, and 0.02 N/mm^2^ at 400 Hz and 1 kHz frequencies, and the common interaction area between the sample and the sound wave “S”, corresponding to the sample’s surface area.

Furthermore, electric power is calculated through the integration of voltage, following the equation:(9)$$P=\frac{\int_{t_{1}}^{t_{2}}\frac{U^{2}}{R}\mathrm{dt}}{\Delta t}$$
where “t_1_” and “t_2_” define the integration bounds, representing the signal’s start and end times, “U” signifies the measured voltage signal, “R” is equal to 1 MΩ, signifying the oscilloscope input impedance, and “Δt” represents the time span, calculated as t_2_ − t_1_. The calculated power values for both sine and square acoustic waves are further detailed in Table 4 and Table 5, respectively. It is crucial to highlight that, in this context, the oscilloscope input impedance is 1 MΩ, serving as the load resistance. When comparing this load resistance with the sample resistances determined from the I-V characteristics, as shown in Table 1, it becomes evident that achieving impedance matching would require a load impedance more than one order of magnitude greater. Consequently, one can anticipate higher-output power under conditions of matched impedance.

For comparative analysis, the power generated under the applied force “P_F_” can be computed by substituting the power expression from Equation (9) into Equation (8), replacing the peak-to-peak voltage. 

Additionally, the power volume density, an important parameter for describing the specimen’s properties more precisely, can be evaluated using the following equation:(10)$$P_{V}=\frac{P}{V}$$
where “P” signifies the generated power from Equation (9) and “V” represents the sample volume.

The results stemming from the voltage and electric power calculations for the investigated samples are conveniently summarized in Table 4.

The observed trend illustrates a clear correlation between the generated power and the frequency increase across all examined samples. This phenomenon arises from the fact that elevating the frequency of the stimulating signal directly leads to an amplification in the frequency of the piezoelectric response from the sample. Consequently, the number of peaks occurring per unit of time rises, directly impacting the overall power output. Surprisingly, it is evident that pure PVDF generates higher voltage and power compared to PVDF with a 5 wt% addition of SbSI. Moreover, with a further increase in SbSI concentration, the signals for composite materials, in general, exhibit slightly superior performance compared to the pure PVDF sample. This outcome finds an explanation in the fundamental physical phenomena previously discussed (Figure 7). The weak piezoelectric effect inherent in PVDF generates charges, which could be captured by SbSI dipoles. As SbSI concentration increases, the piezoelectric effect originating from SbSI nanowires becomes dominant.

Upon analyzing the peak-to-peak voltage against frequency for all samples, an interesting trend emerges. In the case of pure PVDF, the voltage response exhibits continuous growth as frequency increases. For the composite containing 5%wt. of SbSI, the highest generated voltage occurs at 200 Hz, while for 10%wt. and 15%wt. of SbSI, the peak voltage is observed at 400 Hz. This observation suggests the presence of a resonance frequency within this range for the composite samples. The shift in these frequencies with varying SbSI concentrations could be attributed to both the change in SbSI concentration and the alteration in the dimensions of PVDF grains (Figure 3f). Both these factors impact the mechanical properties of the samples and, subsequently, the resonant frequency.

The pure PVDF sample exhibits a consistent response across all measured frequencies. In the case of composite samples, a stable piezoelectric voltage response is observed at 100 and 200 Hz. However, there is evidence of some interfering influences leading to a humming phenomenon at frequencies of 400 Hz and 1 kHz (Figure 8). For composites containing 10%wt. and 15%wt. of SbSI, a clear lack of response stability is evident, depicted by relatively significant voltage peaks. The amplitudes of these peaks are also affected by distinct signal-to-noise ratios (SNRs), attributed to the varied impedance properties of the tested materials [55]. Additionally, this phenomenon results from the differing mechanical properties, which are visibly discernible in Figure 9, particularly at the lowest excitation frequency of 100 Hz. It is important to note that only a subset of the observed peaks corresponds to genuine piezoelectric responses due to the potential presence of interference and humming. In Figure 9, a comprehensive analysis of piezoelectric voltage responses is presented for samples stimulated acoustically using both sine and square waveforms at frequencies of 100 Hz and 1 kHz. These analyses cover three consecutive signal periods. Interestingly, it is observed that at low excitation frequencies, the number of peaks exceeds three, indicating a more complex response pattern.

A single-frequency sine wave was chosen for vibration simulation due to its straightforward description and ease of analysis. On the other hand, the selection of a square wave was motivated by the desire to simulate pressure scenarios involving multiple frequencies simultaneously, closely resembling real-world applications. Sine vibration serves as a fundamental testing method to portray an element’s behavior under vibration conditions. Its primary advantage lies in the simplicity of process control, attributed to the presence of a single sine wave component. In contrast, square waves encompass a fundamental frequency along with an infinite series of odd harmonics, thereby simulating responses that closely resemble real-world applications. This comprehensive approach aids in capturing intricate responses observed in practical scenarios.

Based on the acquired signal, a similar calculation was conducted as in the case of sine excitation to determine peak-to-peak voltage, power (Equation (9)), and volumetric power density (Equation (10)). However, for square signals, the calculations relating to the applied force were not performed due to the presence of an infinite number of odd harmonics in the excitation signal, leading to inaccuracies in force calculation. The results for all the recorded frequencies are presented in Table 5.

To provide a comprehensive understanding, these results can be compared to those obtained under sine excitations, as presented in Table 4. The comparisons highlight distinct response behaviors under the two excitation waveforms. Generally, square wave excitation tends to result in higher peak-to-peak voltages and power outputs for specific compositions and frequencies. 

This enhancement is attributed to the piezotronic properties, and the mechanical stress induced on SbSI nanowires [56]. In the case of a square signal, this stress arises not only from the primary excitation frequency, but also from higher harmonics. Furthermore, at higher frequencies, the process of discharging the matrix becomes apparent. Figure 9b,d displays the drift effect in voltage response values in relation to the capacitor’s charge. This phenomenon is noticeable at higher frequencies, while at lower frequencies, only noise is discernible. Overall, other parameters and variations among the samples align in a general sense with the discussions outlined earlier in Figure 8. However, these aspects are notably influenced by the presence of higher harmonics.

Comparatively, the tested samples exhibit lower responses compared to conventional piezoelectric and piezotronic materials [57,58]. However, a distinct advantage lies in the material’s heightened flexibility, a feature substantiated by current–voltage measurements. It is important to acknowledge that the presented results align with those of other nanocomposites. It should be noted that the tests were conducted under relatively low-force conditions [59]. For a comprehensive evaluation, the generated power was juxtaposed with that of other nanocomposites and MEMS devices, as illustrated in Table 6.

In the provided table, a comprehensive comparison of power outputs is presented for the PVDF/SbSI composite alongside other composite materials, all stimulated by acoustic waves. Additionally, the table includes instances where either PVDF or SbSI is a constituent component. This comparative analysis sheds light on the distinctive power-generation capabilities of the PVDF/SbSI composite in comparison to other related composites, offering insights into their potential applications within energy-harvesting systems.

One can observe that the generated power from our device is lower compared to the PZT metamaterial [62] and PVDF array [63], both of which are also excited by sound waves. Nevertheless, when considering the dimensions of these devices (for instance, the PVDF array has a volume nearly 10,000 times larger than our device), the power generated per unit of device volume would actually be higher in the case of the nanocomposite described here. Importantly, it should be emphasized that while [60,61,62,63] provide device dimensions for the entire apparatus, our study focuses on the active area of the generator due to the distinct construction of devices. Consequently, we have refrained from calculating and comparing the volume density of power in these cases.

Upon comparing our nanocomposite with other SbSI-based nanocomposites, it becomes apparent that both the generated power and power density are greater for the new nanocomposite. However, it is worth noting that the speaker employed in the studies [34,35] was smaller than the one used in our work, resulting in lower acting sound pressure in those cases. Additionally, the measurement of acting force was not conducted in [34,35], preventing a direct comparison. Notably, the achieved power density in our study is approximately 10 times higher than that observed in the fiber-type PVDF/SbSI nanocomposite excited by vibrations [28]. Even though the nanocomposite in that study was a 1-3 type [44] with aligned SbSI nanowires, it is essential to acknowledge that vibration-based excitation might have a less pronounced effect on SbSI nanowire deformation within the composite compared to the actual deflection caused by sound waves.

In a broader context, it is evident that the nanocomposite presented in this research holds promise as a suitable material for constructing nanogenerators.

## 5. Conclusions

A nanocomposite piezotronic sensor was fabricated using a simple casting method. The presented process is both simple and scalable, allowing one to produce a fully assembled system within a few hours, encompassing the production of individual materials and the subsequent mixing of nanocomposites. 

This study delved deeply into the piezoelectric and piezotronic properties of the PVDF/SbSI nanocomposite, yielding valuable insights into its potential for energy-harvesting applications. SEM and EDS investigations were conducted, encompassing current–voltage characteristics and the piezoelectric responses to acoustic wave excitation. The findings underscore the intricate interplay between material composition, mechanical strain, and electrical response, illuminating the underlying mechanisms that govern the composite’s behavior. The physical basis of this behavior was proposed, taking note of the influence of SbSI nanowires on the PVDF matrix structure within the nanocomposite. As the SbSI content percentage increases, there is a corresponding enlargement in the PVDF grain size.

The impact of strain on current–voltage characteristics was aptly demonstrated, with the extent of change in characteristics being contingent on the % SbSI content. This phenomenon becomes more pronounced with increasing SbSI concentration, affirming the existence of the piezotronic effect in the SbSI nanocomposite. This research substantiates the primary influence of nanowires on the piezotronic effect within the composite. The exponential rise in electrical resistivity with increasing SbSI content emphasizes the significance of the nanowires’ piezoelectric and piezotronic attributes in shaping the composite’s electrical behavior. The observable positive pressure coefficient of resistance (PPCR) effect, linked to the nanowires’ high aspect ratio, showcases the composite’s distinct ability to modulate resistance in response to external mechanical forces. This behavior holds promise for applications such as strain sensing, self-powered sensors, and memristors.

Dynamic strain testing has revealed the potential of the PVDF/SbSI nanocomposite as a piezoelectric nanogenerator (PENG). By efficiently converting mechanical energy into electrical energy, the composite demonstrates its ability to harness sound wave vibrations and mechanical deformations for power generation. The presented comparative analysis against other materials and configurations underscores the competitive power density of the composite, making it an attractive candidate for energy harvesting in various scenarios.

In the broader context of sustainable energy solutions, the PVDF/SbSI nanocomposite holds great promise. Its flexible and tunable properties, coupled with its efficient energy-conversion capabilities, position it as an asset in the quest for harnessing ambient energy sources to power electronic devices and sensor networks. As we continue to advance our understanding of nanomaterials and their unique interactions, the path towards self-powered and environmentally friendly technologies becomes clearer, paving the way for a more energy-efficient future.

Looking ahead, this research opens avenues for further exploration and refinement. The study of piezoelectric and piezotronic effects in nanocomposites could lead to the development of innovative materials and devices with enhanced performance characteristics. Future research could focus on optimizing the composite’s composition, nanostructure, and processing techniques to achieve even higher power-generation efficiencies. Moreover, investigating the composite’s long-term stability, scalability, and integration into practical energy-harvesting systems is crucial for real-world applications.

## Figures and Tables

**Figure 1 sensors-23-07855-f001:**
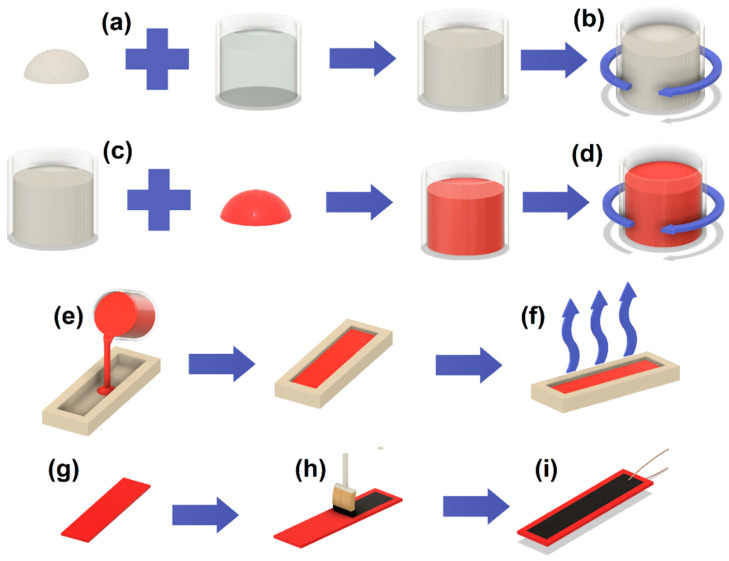
Scheme of sample preparation: (**a**) adding PVDF to DMF, (**b**) stirring of the mixture until PVDF dissolves, (**c**) adding SbSI nanowires to the solution, (**d**) stirring the mixture until homogeneity is achieved, (**e**) casting DMF PVDF/SbSI mixture into a silicone mold, (**f**) evaporating DMF at an elevated temperature (80 °C), (**g**) sample after DMF evaporation removed from the silicon mold, (**h**) painting electrodes on both sides of the sample using conductive carbon paint, (**i**) establishing a connection with copper wires before conductive paint dries; finished sample.

**Figure 2 sensors-23-07855-f002:**
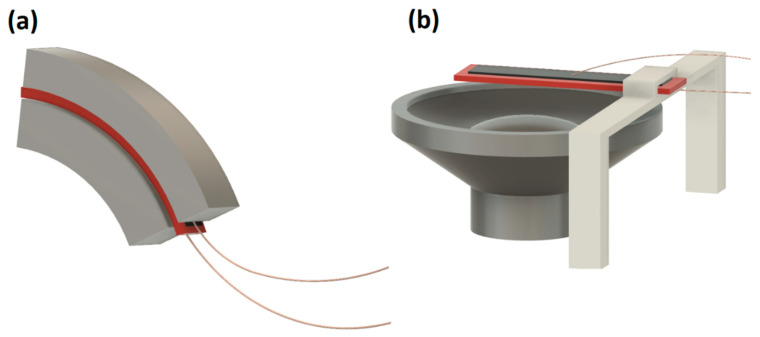
Schematic Representation of (**a**) static testing, in which the sample is subjected to deflection between arc-shaped elements, and (**b**) dynamic testing, with the sample positioned above a loudspeaker.

**Figure 3 sensors-23-07855-f003:**
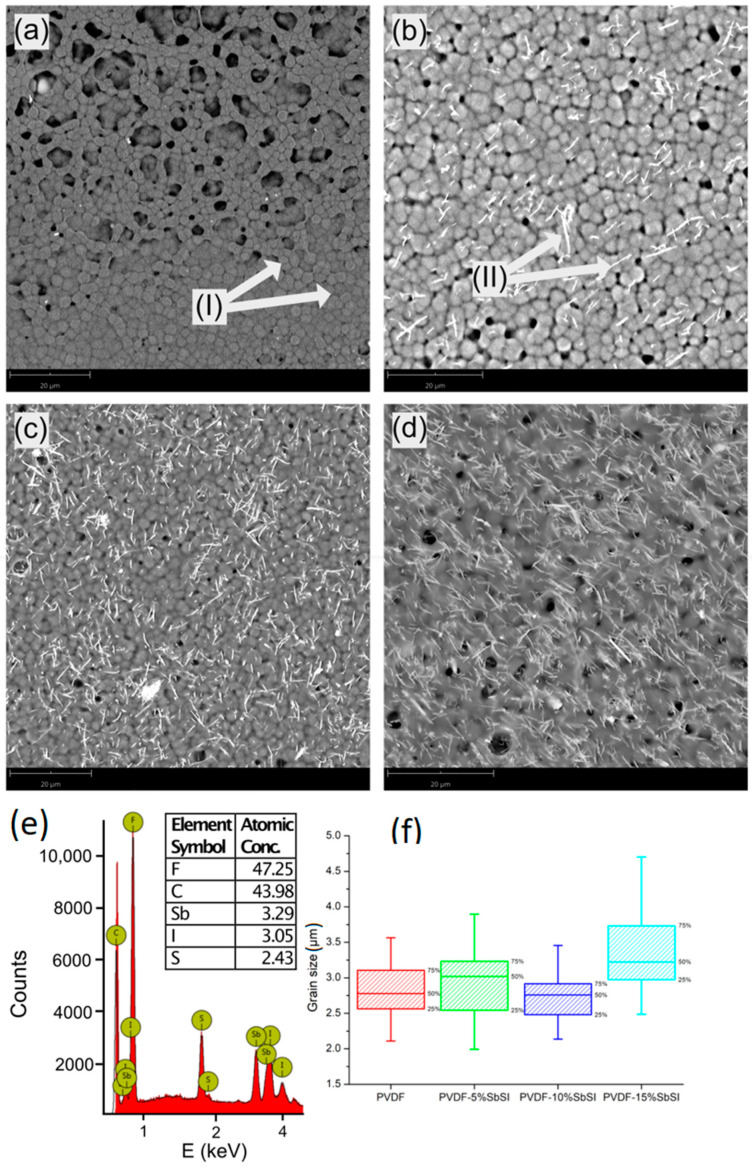
SEM micrographs of various samples: (**a**) pure PVDF, where (I) indicates the PVDF grains; (**b**) PVDF with 5% weight SbSI, where (II) depicts the SbSI nanowires; (**c**) PVDF with 10% weight SbSI; (**d**) PVDF with 15% weight SbSI. (**e**) The EDS spectrum for the sample containing 10% weight SbSI; the inset table indicates the atomic concentration of the nanocomposite. (**f**) Distribution of PVDF grain sizes in the samples.

**Figure 4 sensors-23-07855-f004:**
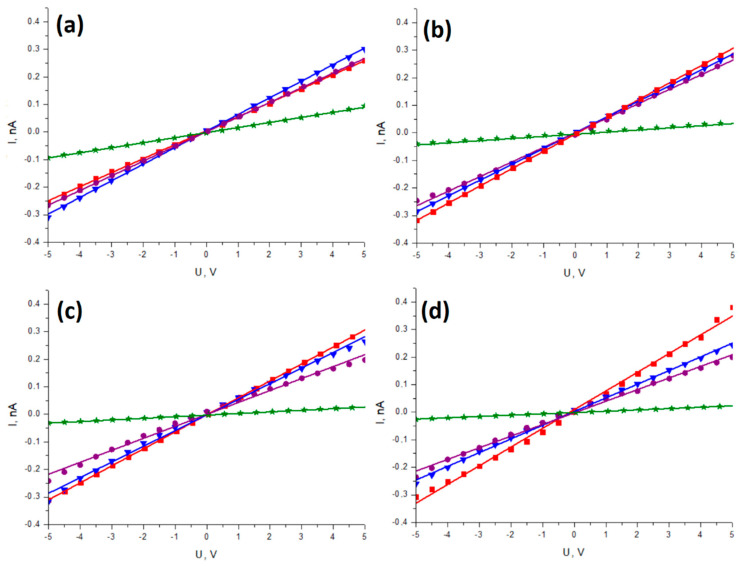
I–V characteristics under different static deflections: (**a**) 0°, (**b**) 15°, (**c**) 30°, (**d**) 60°, registered for pure PVDF (▮), as well as PVDF composite containing 5% wt. SbSI (▼), 10% wt. SbSI (●), and 15% wt. SbSI (★).

**Figure 5 sensors-23-07855-f005:**
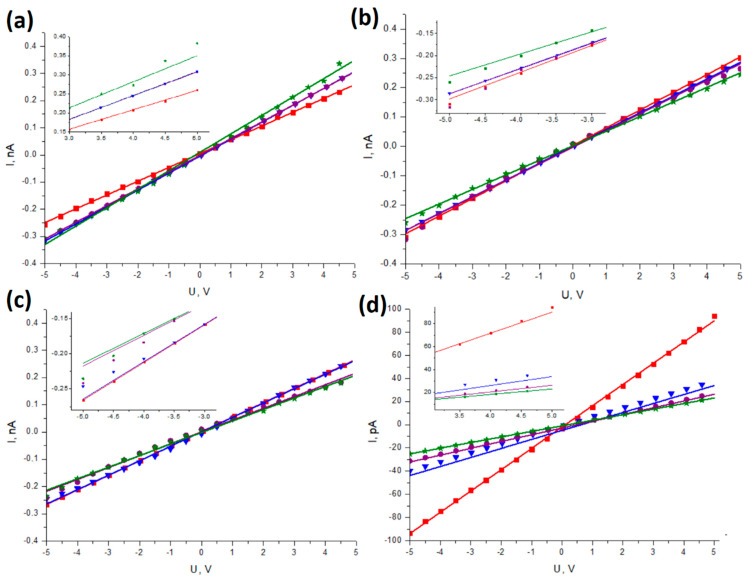
I–V characteristics of pure PVDF (**a**), as well as PVDF composite containing 5% wt. SbSI (**b**), 10% wt. SbSI (**c**), and 15% wt. SbSI (**d**) registered under different static deflection angles: (▮) 0°, (▼) 15°, (●) 30°, and (★) 60°. Insets present part of the graphs to increase visibility of the indicated nonlinearities.

**Figure 6 sensors-23-07855-f006:**
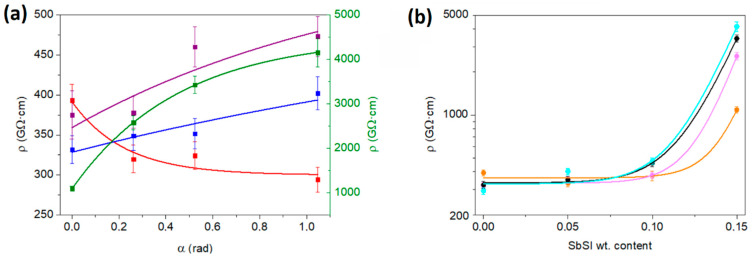
(**a**) The electrical resistivity of PVDF (▮), PVDF with 5% wt. SbSI (▮), and PVDF with 10% wt. SbSI (▮) as a function of sample deflection angle in radians plotted on the black left-side scale. PVDF with 15% wt. SbSI (▮) is plotted on the green right-side scale. (**b**) Electrical resistivity plotted against the weight content of SbSI nanowires for the undeflected nanocomposite sample (●), and nanocomposite sample deflected by 15 (●), 30 (●), and 60 (●) degrees. Error bars correspond to calculated uncertainties. Curves present fitted empirical dependences. Description in the text.

**Figure 7 sensors-23-07855-f007:**
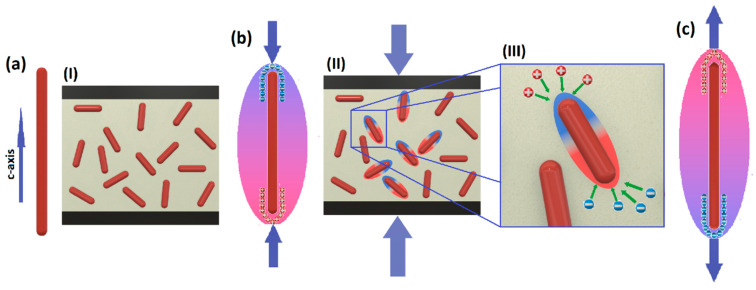
Schematic representations of (**a**) unstrained SbSI nanowire and (I) unstrained nanocomposite sample with SbSI nanowires embedded in a PVDF matrix, sandwiched between carbon-based electrodes. (**b**) The piezopotential generation process in compressed SbSI nanowires, along with a visual representation (II) of the nanocomposite under strain, in which SbSI nanowires act as electric dipoles, (III) attracting charge carriers with the opposite sign towards them. (**c**) Elongated SbSI nanowires with reversed piezoelectric potential.

**Figure 8 sensors-23-07855-f008:**
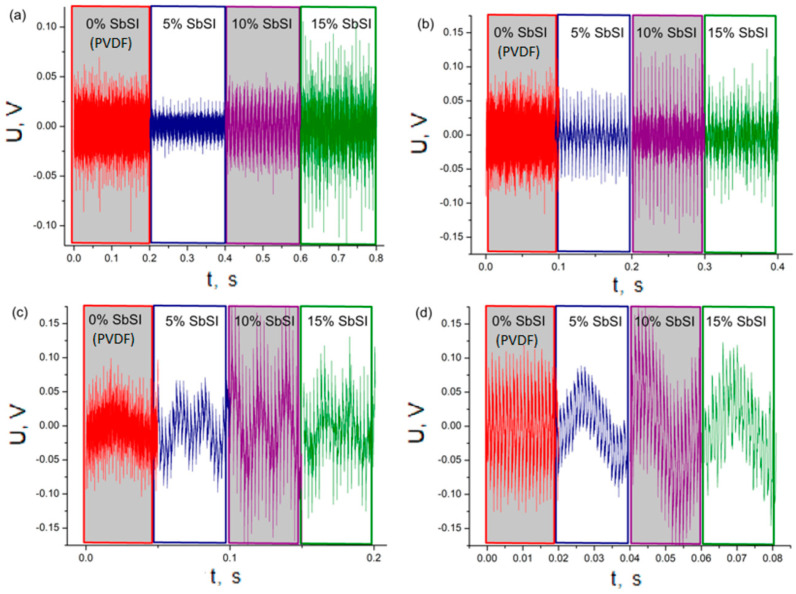
Voltage−driven piezoelectric response of samples with varying weight concentrations of SbSI when subjected to acoustic sine waves at specific frequencies: (**a**) 100 Hz, (**b**) 200 Hz, (**c**) 400 Hz, and (**d**) 1 kHz.

**Figure 9 sensors-23-07855-f009:**
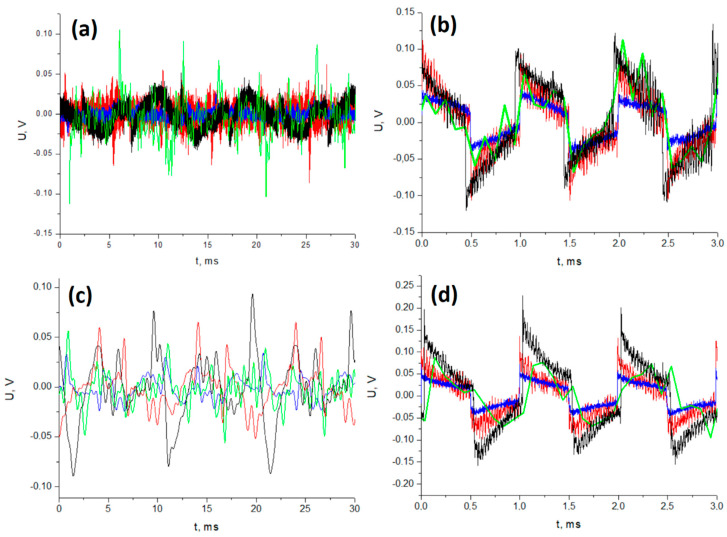
Comparative analysis of the piezoelectric voltage responses of samples under acoustic excitation using sine (**a**,**b**) and square (**c**,**d**) waveforms, at frequencies of 100 Hz (**a**,**c**) and 1 kHz (**b**,**d**) for the following samples: ― pure PVDF; ― PVDF with 5%wt. of SbSI; ― PVDF with 10%wt. of SbSI; and ― PVDF with 15%wt. of SbSI.

**Table 1 sensors-23-07855-t001:** The fitting parameters of the I-V dependencies (Figure 4 and Figure 5), along with the calculated resistance of the sample and electrical resistivity of the nanocomposite for various deflection angles and SbSI contents.

Sample	Deflection, ⁰	a, pA/V	b, pA	R, MΩ	ρ, GΩ·cm
Pure PVDF	0	50.85 (24)	4.80 (73)	19.665 (30)	393 (20)
	15	62.5 (11)	−4.7 (34)	15.99 (29)	319 (17)
	30	61.7 (11)	−1.7 (34)	16.21 (30)	324 (18)
	60	67.97 (98)	9.6 (30)	14.71 (21)	294 (16)
5% wt. SbSI	0	60.31 (31)	3.60 (94)	16.581 (85)	331 (17)
	15	57.30 (51)	0.5 (16)	17.45 (16)	349 (18)
	30	56.87 (86)	−2.1 (26)	17.58 (27)	351 (19)
	60	49.71 (40)	2.5 (12)	20.12 (16)	402 (21)
10% wt. SbSI	0	53.3 (33)	0.8 (99)	18.8 (12)	375 (30)
	15	52.93 (58)	0.4 (18)	18.89 (21)	378 (20)
	30	43.46 (80)	−0.1 (24)	23.01 (42)	460 (25)
	60	42.26 (61)	−2.1 (19)	23.66 (34)	473 (25)
15% wt. SbSI	0	18.39 (10)	−1.98 (31)	54.36 (30)	1087 (56)
	15	7.75 (27)	−4.95 (81)	128.9 (44)	2577 (158)
	30	5.83 (13)	−2.86 (38)	171.5 (37)	3429 (190)
	60	4.81 (29)	−0.99 (87)	208 (12)	4155 (326)

**Table 2 sensors-23-07855-t002:** The fitting parameters of the electrical resistivities vs. deflection angle as presented in Figure 6a. Description in the text.

Sample	a_1_, GΩ·cm	b_1_, GΩ·cm	c_1_
Pure PVDF	300 (18)	−92 (27)	0.011 (38)
5% wt. SbSI	533.6 (84)	205.1 (83)	0.694 (61)
10% wt. SbSI	577 (29)	218 (28)	0.46 (20)
15% wt. SbSI	4504 (26)	3417 (26)	0.1110 (35)

**Table 3 sensors-23-07855-t003:** The fitting parameters of the electrical resistivities vs. the weight content of SbSI as presented in Figure 6b. Description in the text.

Deflection, ⁰	ρ_0_, GΩ·cm	a_2_, GΩ·cm	b_2_
0	363 (31)	0.003 (39)	0.012 (14)
15	333 (14)	0.019 (33)	0.0128 (20)
30	334 (12)	0.22 (13)	0.0157 (10)
60	329 (51)	0.24 (57)	0.0155 (40)

**Table 4 sensors-23-07855-t004:** Measurement results of peak-to-peak voltage “U_p-p_”, force-related peak-to-peak voltage “U_p-pF_”, generated power “P”, generated power under applied force “P_F_”, and power volume density “P_V_” for investigated samples excited by a sine acoustic wave at different frequencies. The colors in the table rows correspond to the signal colors in Figure 8 and serve to enhance the clarity of the results.

f, Hz	Sample	U_p-p_, V	U_p-pF_, V/N	P, nW	P_F_, nW/N	P_V_, mW/m^3^
100	Pure PVDF	0.1148 (74)	0.0163 (11)	0.238	0.034	1.19
	5% wt. SbSI	0.0489 (23)	0.00693 (32)	0.0364	0.005	0.182
	10% wt. SbSI	0.0853 (27)	0.01209 (39)	0.307	0.044	1.54
	15% wt. SbSI	0.1705 (99)	0.0242 (14)	0.725	0.103	3.63
200	Pure PVDF	0.1583 (44)	0.01908 (53)	0.553	0.067	2.76
	5% wt. SbSI	0.1203 (32)	0.01450 (38)	0.437	0.053	2.18
	10% wt. SbSI	0.2232 (79)	0.02690 (95)	0.882	0.106	4.41
	15% wt. SbSI	0.154 (11)	0.0218 (15)	1.27	0.153	6.33
400	Pure PVDF	0.1501 (85)	0.0187 (11)	0.611	0.076	3.05
	5% wt. SbSI	0.099 (13)	0.0123 (16)	1.17	0.146	5.86
	10% wt. SbSI	0.267 (22)	0.0334 (28)	3.43	0.429	17.2
	15% wt. SbSI	0.176 (15)	0.0220 (19)	1.82	0.228	9.11
1000	Pure PVDF	0.2156 (59)	0.02696 (74)	2.14	0.267	10.7
	5% wt. SbSI	0.089 (16)	0.0112 (20)	1.50	0.188	7.51
	10% wt. SbSI	0.215 (21)	0.0269 (27)	6.71	0.838	33.5
	15% wt. SbSI	0.140 (16)	0.0174 (20)	4.10	0.513	20.5

**Table 5 sensors-23-07855-t005:** Measurement results of peak-to-peak voltage “U_p-p_”, generated power “P”, and power volume density “P_V_” for investigated samples excited by a square acoustic wave at different frequencies. The colors in the table rows correspond to the signal colors in Figure 9 and serve to enhance the clarity of the results.

Frequency, Hz	Sample	U_p-p_, V	P, nW	P_V_, mW/m^3^
100	Pure PVDF	0.1128 (39)	0.516	2.58
	5% wt. SbSI	0.0537 (14)	0.134	0.67
	10% wt. SbSI	0.1697 (42)	1.12	5.62
	15% wt. SbSI	0.0910 (63)	0.36	1.8
200	Pure PVDF	0.207 (10)	0.879	4.4
	5% wt. SbSI	0.1885 (59)	0.704	3.52
	10% wt. SbSI	0.1094 (34)	0.446	2.23
	15% wt. SbSI	0.1230 (66)	1.08	5.4
400	Pure PVDF	0.2118 (73)	1.20	5.99
	5% wt. SbSI	0.0829 (33)	0.099	0.50
	10% wt. SbSI	0.1109 (44)	0.532	2.66
	15% wt. SbSI	0.1385 (78)	1.26	6.32
1000	Pure PVDF	0.210 (11)	2.16	10.8
	5% wt. SbSI	0.1132 (33)	0.831	4.16
	10% wt. SbSI	0.3349 (66)	6.44	32.2
	15% wt. SbSI	0.169 (13)	2.14	10.7

**Table 6 sensors-23-07855-t006:** Comparison of power outputs for various composites that are stimulated by acoustic waves, and composites that incorporate PVDF or SbSI and are stimulated by factors beyond acoustic waves.

Material	Common Factor	Device Size, cm^3^	P, nW	P_V_, mW/m^3^	Parameters	Ref.
PZT (MEMS)	Sound wave excitation	2.5	0.006	-	The sound pressure level of 149 dB, resonant frequency of 3.9 kHz	[60]
PDMS/CNT	Sound wave excitation	6.9	40	-	An acoustic wave of 4.24 kHz, with different power inputs to the speaker (80, 40, and 20 W)	[61]
PZT (metamaterial)	Sound wave excitation	676	345	-	The sound pressure level of 100 dB at 600 Hz of frequency	[62]
PVDF (beam array)	Sound wave excitation and PVDF material	1500	2200	-	The sound pressure level of 100 dB, frequencies of 146, 439, and 734 Hz	[63]
Epoxy/SbSI	Sound wave excitation and SbSI-based nanocomposite	2	0.9	0.45	An acoustic wave of 175 Hz frequency, sound pressure level of 90 dB	[34]
PVDF/SbSI (fibers)	PVDF material and SbSI-based nanocomposite	0.005	0.017	3.46	Mechanical vibrations of 50 Hz frequency	[28]
Cellulose/SbSI	Sound wave excitation and SbSI-based nanocomposite	0.005	0.208	41.5	An acoustic wave of 175 Hz frequency, sound pressure level of 90 dB	[35]
PVDF/SbSI	This paper	0.2	6.71	33.5	An acoustic sine wave of frequency 1 kHz, Force F = 8.3 N	This paper

## Data Availability

Data available upon request.

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
