# Peer review of "Piezotronic Antimony Sulphoiodide/Polyvinylidene Composite for Strain-Sensing and Energy-Harvesting Applications"

_sensors, 2023, doi:10.3390/s23187855_

Round 1

Reviewer 1 Report

In the manuscript proposed by Jala et al, the piezoelectric properties of a PVDF- and SbSI-based composite are investigated, trying to understand what is the effect of the concentration ratio of SbSI in order to optimize the response of the material itself to mechanical strain for possible applications in the Piezotronic field. The paper is well written and extensively argued, and I really find the topic interesting, as well as the authors' proposal of composite material. Some points in the manuscript are in my opinion to be revised in order to make it more organic and scientifically relevant.
In detail:
(a) The introduction is very well discussed and argued, and the bibliographical citations are also adequate.
(b) Sample preparation: figure #1 does not seem useful in describing the process of sample making, being sufficiently described in the text, and offering no additional details to the explanation. I suggest that it be removed. In contrast, line 108 refers to synthesis only through citation: I suggest that some brief indication of the synthesis process be given in this manuscript as well. Instead, a figure or picture of the arc-shaped elements could be very helpful to better understand how they are made.
(c) SEM Examination: the SEM analysis is not clear and properly described. First, it is not specified whether the SEM images presented are of a cross section or of the sample surface. Also lacking is an adequate description of how the analysis of the PVDF grain size variation was performed: how was the grain size calculated? From the graph in Fig. 2i, there is not much variation as reported in the text, except perhaps for the 15% SbSI content for which it is a bit more noticeable. The reported reference ([45]) does not seem to entirely satisfy the behavior observed by the authors for the PVDF-SbSI case.
Finally, the EDS analysis is not very convincing: was the stoichiometric ratio done with or without a reference of a SbSI standard? What kind of decomposition do the authors expect from the introduction of SbSI in PVDF? And due to what?
(d) Characterization of samples under static deformation: the characterization is clearly and precisely described. Only in Fig. 4b and c the deviation from the linear trend as reported in the tense is not clear. I suggest a better presentation of the graphs (perhaps by reporting an enlargement of the higher-V area). Regarding graphs 5a and 5b: why are the values given in radians and not in degrees as on the text? Using the same colors to identify different measurements makes the two graphs very confusing. Also unclear is the explanation in paragraph 289-293: why should increasing the SbSI content increase the resistivity? By the way, it appears from the graph that resistivity also increases for 15% content, contrary to what is perceived in the text (line 291).
(e) Dynamic strain tests: This paragraph is very precisely and clearly written, including the comparison with the other composites given in table 6. I suggest again to revise the colors of the graphs, in fig.8.
(f) Conclusions: to be revised as the text changes.

Reviewer 2 Report

This work presents a novel composite material comprising polyvinylidene fluoride (PVDF) and antimony sulphoiodide (SbSI) nanowires. The material’s electrical resistivity, piezoelectric response, and energy harvesting capabilities are systematically analysed under various deflection conditions and excitation frequencies. The manuscript is well written and data presented is reasonable, so I think it can be published after minor revisions as below:

1.   Please add some related references in the Introduction, such as the applications of the strain sensors in human motion sensing “Hong Y, Wang B, Lin W, et al. Highly anisotropic and flexible piezoceramic kirigami for preventing joint disorders[J]. Science advances, 2021, 7(11): eabf0795.” and acoustic sensing “Jung Y H, Hong S K, Wang H S, et al. Flexible piezoelectric acoustic sensors and machine learning for speech processing[J]. Advanced Materials, 2020, 32(35): 1904020.”

2.  In Figure 2, you mentioned the aggregation of the nanowires. So it is a good thing or a bad thing to the material properties?

3.  It’s better to use different patterns in the legend such as triangle, circle and diamond, instead only the square pattern in Figure 3, 4, and 5.

4.  In lines 289-290, the content increase of the material with a lower resistance (SbSI) should leads to a resistance decrease. So why you claim “the resistivity generally increases with the weight content of SbSI”?

5.   What’s the piezoelectric constant d33 of the composites?

6. In Table 4 and 5, how you calculate the power output? What’s the matched impedance here?

7.  Please elaborate the novelty of this work compared with the previously reported works.

Minor editing of English language required
